# Natural and Synthetic Estrogens in Chronic Inflammation and Breast Cancer

**DOI:** 10.3390/cancers14010206

**Published:** 2021-12-31

**Authors:** Chandra K. Maharjan, Jiao Mo, Lei Wang, Myung-Chul Kim, Sameul Wang, Nicholas Borcherding, Praveen Vikas, Weizhou Zhang

**Affiliations:** 1Department of Pathology, Immunology and Laboratory Medicine, University of Florida, Gainesville, FL 32610, USA; cmaharjan@ufl.edu (C.K.M.); mojiao@ufl.edu (J.M.); lei.wang@ufl.edu (L.W.); my.kim@ufl.edu (M.-C.K.); 2Canyonoak Consulting LLC, San Diego, CA 92127, USA; samuelwang@canyonoak.org; 3Department of Pathology and Immunology, School of Medicine, Washington University, St. Louis, MO 63110, USA; borcherding.n@email.wustl.edu; 4Department of Internal Medicine, Carver College of Medicine, Iowa City, IA 52242, USA; Praveen-vikas@uiowa.edu; 5Mechanism of Oncogenesis Program, University of Florida Health Cancer Center, University of Florida, Gainesville, FL 32610, USA

**Keywords:** natural estrogens, synthetic estrogens, environmental estrogens, xenoestrogens, chronic inflammation, breast cancer

## Abstract

**Simple Summary:**

Estrogens regulate key physiological functions in the human body, including the development of female reproductive organs. However, dysregulated estrogen signaling—mainly mediated by estrogen receptors (ER)—is associated with many developmental, mental, and other diseases, including cancer. An increasing number of studies have demonstrated that estrogen modulates inflammatory processes within many tissues. In addition to estrogens made within the body, humans are also constantly exposed to many outside estrogens, naturally occurring in plants and those artificially synthesized in industries. Here, we will also discuss the link between chronic exposure to environmental estrogens with tissue inflammation and breast cancer development.

**Abstract:**

The oncogenic role of estrogen receptor (ER) signaling in breast cancer has long been established. Interaction of estrogen with estrogen receptor (ER) in the nucleus activates genomic pathways of estrogen signaling. In contrast, estrogen interaction with the cell membrane-bound G-protein-coupled estrogen receptor (GPER) activates the rapid receptor-mediated signaling transduction cascades. Aberrant estrogen signaling enhances mammary epithelial cell proliferation, survival, and angiogenesis, hence is an important step towards breast cancer initiation and progression. Meanwhile, a growing number of studies also provide evidence for estrogen’s pro- or anti-inflammatory roles. As other articles in this issue cover classic ER and GPER signaling mediated by estrogen, this review will discuss the crucial mechanisms by which estrogen signaling influences chronic inflammation and how that is involved in breast cancer. Xenoestrogens acquired from plant diet or exposure to industrial products constantly interact with and alter innate estrogen signaling at various levels. As such, they can modulate chronic inflammation and breast cancer development. Natural xenoestrogens generally have anti-inflammatory properties, which is consistent with their chemoprotective role in breast cancer. In contrast, synthetic xenoestrogens are proinflammatory and carcinogenic compounds that can increase the risk of breast cancer. This article also highlights important xenoestrogens with a particular focus on their role in inflammation and breast cancer. Improved understanding of the complex relationship between estrogens, inflammation, and breast cancer will guide clinical research on agents that could advance breast cancer prevention and therapy.

## 1. Introduction

Estrogen sex steroid hormones orchestrate a multitude of physiological functions ranging from development and regulation of the human reproductive system to modulation of neuroendocrine, skeletal, adipose, and cardiovascular systems [1,2]. Endogenous estrogens in humans include 17 β-estradiol (E2), estrone (E1), and estriol (E3), with E2 being the predominant type. E1 and E2 are primarily secreted by the ovaries and at low levels by the adipose tissue and adrenal glands in premenopausal women. E3, mainly secreted by the placenta, is the dominant estrogen during pregnancy [3]. In males, testes produce only one-fifth of circulating estrogens, with extragonadal organs, including adipose, brain, skin, and bone, producing the remainder [4]. The hypothalamic–pituitary–ovarian hormonal axis regulates ovarian estrogen production. The hypothalamic peptide gonadotropin-releasing hormone (GnRH) stimulates the luteinizing hormone (LH) and the follicle-stimulating hormone (FSH) release from the anterior pituitary. LH stimulates androgen production, whereas FSH works in concert with LH to upregulate aromatase, a cytochrome P450 enzyme that catalyzes androgen to estrogen [5]. High levels of E2 production during ovulation limit the release of GnRH, LH, and FSH via a negative feedback mechanism.

Although endogenous estrogens are physiologically essential, aberrant ER signaling enhances the risk of cancer development [2]. In 1896, George Beatson demonstrated the remission of breast cancer following bilateral oophorectomy in premenopausal women, pioneering the connection between estrogen/ER and breast cancer [6]. Since then, a plethora of evidence has further substantiated the oncogenic role of both endogenous and exogenous estrogens in the development and progression of various cancers, primarily those of the breast and endometrium, as well as those of the ovary, prostate, and lung [7,8]. Estrogens promote tumorigenesis by signaling through genomic and nongenomic pathways. Notably, a growing body of literature has highlighted the role of estrogen in modulating tissue inflammation, which has been summarized in a review [9]. Interestingly, the role of estrogen in inflammation seems to be tissue/disease-dependent. For example, women have a higher rate of autoimmune diseases, comprising 80% of the total autoimmune patients, which is generally attributed to estrogen’s proinflammatory role. In contrast, the role of estrogen as an anti-inflammatory agent has also been documented in several cancers, including that of the liver (incidence ratio between men versus women ranges from 2:1 to 4:1), where estrogen inhibits the production of the inflammatory cytokine interleukin 6 (IL-6) [10]. The anti-inflammatory roles of estrogen have additionally been demonstrated in other models and human diseases [9]. In general, it is well-appreciated that chronic tissue inflammation is an established risk factor for various events of cancer initiation and progression, including cellular transformation, survival, proliferation, invasion, angiogenesis, and metastasis [11,12,13]. In this current article, we review the implications of natural and synthetic estrogens in chronic inflammation and ensuing carcinogenesis.

## 2. Estrogen Signaling

Estrogens exert their biological functions primarily by binding to estrogen receptor (ER) α or ERβ [14]. ERα and ERβ are encoded by two distinct genes, *ESR1* and *ESR2*, respectively, which possess unique, yet overlapping tissue expression profiles [2]. ERs exist as homo- or heterodimers at the peri-membrane, mitochondria, and nucleus [14,15,16,17].

### 2.1. Canonical Estrogen Signaling within the Nucleus

Estrogen signaling in the cells is mediated by both genomic and nongenomic pathways (Figure 1). According to the classical concept, the cytoplasmic estrogen–ER complex is translocated into the nucleus, where it functions as a ligand-activated transcription factor to regulate the expression of its target genes. Mechanistically, estrogen binding to the ER ligand-binding domain induces structural transformation of the inactive ER to its functionally active form. This binding enhances the stability of the ER dimer and facilitates its interaction with co-regulatory proteins [18,19]. In certain instances, the activated estrogen–ER complex interacts with estrogen response elements (EREs), constituting a consensus 5′-CGTCAnnnTGACC-3′ palindrome sequence on the promoter of its target genes to transactivate their expression [20]. Alternatively, the estrogen-activated ER can regulate gene expressions via ERE-independent mechanisms. These mechanisms involve direct or indirect interactions of ER, aided by co-regulatory proteins, with other transcription factors such as activator protein 1 (AP1), specificity protein 1 (SP1), nuclear factor-κB (NF-κB), cAMP response element-binding protein (CREB), p53, and signal transduction and activator of transcription 5 (STAT5) [2,14,21].

### 2.2. Noncanonical Estrogen Signaling on the Surface

Interestingly, exposure of target cells to estrogen can also lead to instant responses, including the induction of ionic fluxes and activation of protein kinases within seconds to minutes, which are too rapid to be explained by ER-induced transcriptional regulation [22]. As such, the nongenomic activity of estrogen is primarily mediated by its interaction with a unique, G-protein-coupled receptor (GPR), GPR30, also known as G protein estrogen receptor (GPER) [22,23]. Membrane-bound GPER monomers rapidly dimerize and activate Gα and Gβγ proteins, responsible for releasing classical second messengers such as cAMP [24], inositol phosphate [25], and calcium [26]. Activated G proteins can also stimulate the ERK/MAPK and PI3K/AKT signaling cascades that regulate cell proliferation, migration, and other biological processes [14,22]. Furthermore, GPER-activated kinases have been shown to regulate the genomic functions of estrogen by modifying the nuclear ERs, co-regulatory proteins, and transcription factors at the post-translational level.

### 2.3. Mitochondrial ER Signaling

In mitochondria, estrogen signaling regulates essential physiological functions [27,28]. Both ERα and ERβ have been detected within mitochondria [29] and shown to interact with ERE-like sequences of mouse and human mitochondrial DNA (mtDNA) [30]. This interaction is consistent with the evidence that E2 induces several mtDNA-encoded mRNAs, including the mitochondrial ATP synthase subunit E. Furthermore, the nuclear E2–ERα signaling induces the expression of the nuclear respiratory factor 1 (*NRF-1*) gene from the nuclear DNA [31] and that of NRF-1-regulated nuclear genes such as mitochondrial transcription factor (*TFAM*), mitochondrial transcription factor B1 (*TFB1M*), and *TFB2M*, involved in the transcription of mtDNA-encoded genes, mitochondrial biogenesis, and oxidative phosphorylation [14,31]. Of note, E2–ER binding enhances the expression and activity of manganese superoxide dismutase (Mn-SOD), which scavenges the mitochondrial reactive oxygen species (ROS) and hence, prevents cells from undergoing apoptosis [14].

### 2.4. Differential ERα and ERβ Signaling

The functional consequence of estrogen signaling dramatically depends on the ER subtype engaged during the process. ERα and ERβ share a moderate homology (56%) within their ligand-binding domains, which dictates their differential specificity and affinity towards endogenous and exogenous estrogens as well as antiestrogens [14,32]. Moreover, unlike ERα, ERβ lacks a strong transactivation AF-1 domain, but instead contains a repressor domain [17] that contributes to the ligand-dependent differences in their ability to recruit the transcriptional coactivators [33]. ERβ has been demonstrated to inhibit ERα transcriptional activity at subsaturating concentrations, decreasing the cellular sensitivity to estradiol [17]. It has been found that ERα and ERβ regulate different genes or can differentially regulate the same genes [32]. ERα and ERβ play opposite roles in carcinogenesis, with ERα preferentially acting as an oncogene whereas ERβ acts as a tumor suppressor. ERα promotes tumor cell proliferation and inhibits apoptosis in cancers of the breast, prostate, lung, ovary, and endometrium [32,34]. Exaggerated ERα expression has also been correlated with poor prognosis or drug resistance to endocrine therapies in breast cancer patients. In contrast, ERβ exerts tumor-suppressive functions in most instances. ERβ has been shown to be expressed in the majority of invasive breast cancers including the human epithelial growth factor receptor 2 (HER2) and basal (or triple-negative) subtypes that lack ERα expression [35,36,37]. Although positive ERβ expression is believed to predict better survival and the sensitivity to tamoxifen, contradictory results have also been observed [32,34,37,38,39]. This suggests the role of ERβ in breast cancer is complicated and context-dependent. The observed lack of consistency in the prognostic role of ERβ expression has also been partly attributed to the methodological limitations caused by the unspecific nature of the commercial antibodies employed in previous studies [40]. In sum, the clinical application of ERβ staining as a diagnostic and prognostic marker in breast cancer has remained inconclusive and controversial.

### 2.5. Estrogen Signaling in Breast Cancer

The positive correlation between estrogen/ER activation and breast cancer has long been confirmed in clinical statistics showing that early menarche, late primiparity, and late menopause are risk factors for breast cancers while oophorectomy is correlated with decreased breast cancer incidence [41]. Based on the expression status of hormone receptors (HRs) (estrogen receptor, ER, and progesterone receptor, PR) and human epidermal growth factor receptor 2 (HER2), breast cancer is divided into luminal A (HR+/HER2−, 69%), luminal B (HR+/HER2+, 10%), HER2-enriched (HR−/HER2+, 8%) and triple-negative breast cancers (TNBC) (HR−/HER2−, 13%) [42,43]. Paradoxically, though excessive estrogen/ER activation causes breast cancer, HR-positive breast cancers have an overall positive prognosis and the expression level of ER is the singular good prognosis indicator in breast cancer.

The oncogenic role of estrogens in breast cancer is primarily ERα-dependent. There is plenty of evidence showing the correlation between ERα expression level and the dividing of ER-positive cells in the tumorigenic process [44,45]. It is well established that estrogen–ERα interaction promotes cell cycle progression through the G1-S phase and proliferation by inducing c-Myc and cyclin D1 expression [46,47]. In contrast, ERβ inhibits ERα-induced tumorigenesis in vitro and in vivo with concomitant repression of the cyclin D1, cyclin A, c-Myc, vascular endothelial growth factor (VEGF), and platelet-derived growth factor (PDGF) oncogenes, as well as the induction of p21 and p27 tumor suppressors by ERβ in ER-positive breast cancer cells [34,48,49,50,51]. At the molecular level, ERα, upon binding to estradiol, dimerizes and regulates the transcription of target genes by binding EREs located within their promoter regions [52]. The potential downstream oncogenic mediators of estrogen include the induction of cell cycle promoters cyclin Ds [53], anti-apoptotic BCL family members BCL-2, BCL-W, BCL-X_L_ [54,55,56], and BCL-3 [57], and rapid and membrane-associated signaling events through membrane ER (mER) and intracellular G-protein-coupled estrogen receptor 1 (GPER, GPR30) [58]. Both of these genomic and nongenomic effects of estrogen tend to promote cell proliferation and tumor progression [23,59]. Metastasis is the leading cause of death in advanced breast cancer patients. Co-regulators of ER genomic actions, such as AIB1 (amplified in breast cancer), SRC-1, and PELP1 (proline, glutamate, and leucine-rich protein), have been shown to promote breast cancer metastasis [60]. Interestingly, several studies have shown that ERα can also inhibit breast cancer metastasis. These studies demonstrated that E2–ERα upregulates epithelial marker genes and suppresses mesenchymal marker genes in breast cancer cells, consistent with a decrease in cell migration and invasion [2,61,62]

Interestingly, the consequence of estrogen’s fueling effect is not exclusively a direct proliferation of ER-positive cells in response to estrogen but rather can be mediated by the paracrine influence of ER-positive cells upon the surrounding cells, considering the restricted expression of ER only in 15–30% of luminal cells and the dissociation between ER expression and cell proliferation in mammary tissue [63]. The paracrine-like pattern of estrogen signaling could be the direct adaptation from luminal mammary epithelial cells that, upon hormone activation, leads to the production of WNT ligands and RANKL to induce the activation of HR-negative basal stem cells [64]. Both WNT and RANKL have also been shown to be critical for breast cancer pathogenesis [64,65,66,67]. In the context of inflammation, the paracrine pattern coincides with the tumor microenvironment in which estrogen may interact with inflammatory components, angiogenesis, and lymphangiogenesis to promote tumorigenesis [68].

## 3. Estrogen, Inflammation, and Breast Cancer

The role of estrogen in inflammation is highly complex and context-dependent. Studies show that estrogen can evoke either pro- or anti-inflammatory effects depending on the nature of the immune stimulus, disease stage, target organ, estrogen concentration, and relative expression of ERα versus ERβ [9]. Here, we will discuss the complex interaction between estrogens and inflammation within the context of breast cancer (Figure 2).

### 3.1. Estrogen and Inflammation—The Positive Feedback

Histologically, the breast is composed of hormone-responsive epithelia, stroma, and adipose. Epithelial cells form the mammary duct and subdivide into basal and luminal epithelial cells. Stroma is embedded with fibroblasts and immune cells such as macrophages, which produce and respond to cytokines. Adipose, recognized as an endocrine organ, can secrete hormones, cytokines, and adipokines. In young women, estrogens are produced by the ovary, while in perimenopausal and postmenopausal women at a time when the ovary ceases to synthesize estrogens, the breast can still uptake estrogen from the circulation and synthesize estrogens in situ by the adipose tissue, which constitutes a significant portion of the breast from menopausal women. Such tissue composition predisposes the breast to the occurrence of localized subclinical inflammation [69]. As the obesity epidemic becomes one of the major health concerns in the United States, the connection between obesity and breast cancer has long been established. Adipose tissues from obese women tend to have an increased level of estrogen production. This increased estrogen level from the breast adipose tissues—particularly within obese women—implies an “all weather” impact of estrogens on the breast microenvironment that leads to the increase in breast cancer incidence. Inflammation is a key feature associated with obese adipose tissue [70,71] and could be one of the important contributors to obesity-associated estrogen production. Estrogens are mainly converted from androgens by cytochrome P450 aromatase, encoded by the *CYP19* gene [72]. Aromatase is the convergence point of regulation exerted by inflammatory components. A plethora of studies have evidenced that aromatase activity and its gene transcription are highly susceptible to the regulation of proinflammatory cytokines, resulting in the alterations of estrogen levels upon cytokines’ influence.

#### 3.1.1. IL-6 and Estrogen

IL-6, secreted mainly by immune cells, such as macrophages from obese adipose tissues or breast cancers, stimulates the synthesis of estrogen by activating aromatase as well as enhancing its gene transcription [69,73]. In addition, malignant epithelial cells shed IL-6R, thus giving rise to IL-6 soluble receptors (IL-6 sRs). Estradiol increases IL-6 RNA levels and potentially induces the IL-6 signal transduction, thus forming positive feedback potentiating estrogen and inflammation looping [69]. IL-6 also promotes the conversion of estrogen to estradiol—its biologically active form—by increasing estradiol-17β hydroxysteroid dehydrogenase (17β-HSD) type I. This effect is potentiated by TNF-*α* that increases the expression of IL-6 and the gp130 protein, the signal-transducing component of the IL-6 receptor. IL-1β released by stroma cells and macrophages enhances ER transcriptional activity by displacing the inhibitory complex containing N-CoR (nuclear receptor corepressor) from ER target gene promoters [74]. An in vitro study showed that when administered alone, neither IL-6 nor IL-1β had a significant effect on the growth of ER-positive cell lines, but when combined with estrogen sulfate, IL-6 and IL-1β stimulated cell proliferation markedly by enhancing aromatase and steroid sulfatase activities [75]. Taken together, these proinflammatory cytokines increase the production of estrogen and enhance estrogen signaling, which results in stimulating the proliferation of hormone-dependent breast tumor cells.

#### 3.1.2. PGE2 and Estrogen

Along with the proinflammatory cytokines, inflammatory mediator PGE2 also plays a considerable role in mammary estrogen biosynthesis. PGE2 binds to EP2 receptors to activate PKA pathways and generates cAMP, which stimulates cAMP response element (CRE)-binding protein-1 (CREB1) binding to CREs on aromatase promoter II (PII) and thus enhances the expression of aromatase [76,77,78]. In addition to the direct effect of PGE2 on aromatase expression, a recent study showed that PGE2 increases aromatase by inducing the activating transcription factor 3 (ATF3). ATF3 is a repressive transcription factor that binds to a CREB site within the Sirtuin (*SIRT1*) promoter and reduces SIRT1 levels, while SIRT1 deacetylates and inactivates HIF-1α [79], which is known to enhance aromatase gene expression. The production of PGE2 depends on inducible cyclooxygenase-2 (COX-2), and the mRNA level of *COX2* correlates with the expression of aromatase [80]. A more recent study showed that treatment with a COX-2 selective inhibitor etoricoxib reduced the aromatase protein and estrogen levels and decreased the incidence of mammary tumors in high-sugar/fat-fed mice [81]. In a mouse model of postpartum breast cancer whereby the invasiveness of tumor cells is dependent on COX-2, ibuprofen treatment mitigated the increased tumor growth, decreased lung metastasis, and restored COX-2 expression to the level found in nulliparous counterparts [82]. Other than a unidirectional impact of inflammatory cytokines on estrogen synthesis and ER signaling, estrogen has been revealed to modulate the inflammatory process. At different physiological/developmental stages of the breast, the modulation of estrogen on inflammation varies depending on the stage of mammary development. During mammary involution after weaning, estrogen heightens mammary inflammation in that it facilitates neutrophil infiltration and induces gene expression changes of the inflammatory response pathways. In particular, the expression of the gene encoding COX-2 (*Ptgs2*) is upregulated, while tumor suppressor gene *Rb1* is downregulated [83,84]. The way that estrogen interplays with inflammation cues the transient increased risk of breast cancer postpartum and the poor prognosis of parity-associated breast cancer.

#### 3.1.3. Adipokines and Estrogen

Besides cytokines, adipokines, particularly leptin and adiponectin, are released mostly by adipose tissues interacting with estrogen and ER signaling and thus influence inflammation [85]. Leptin is a pleiotropic protein that regulates a wide range of physiological processes such as food intake, immune responses, and cell proliferation and differentiation [86]. ER is found to be co-expressed with leptin receptor (ObR) in malignant breast tissues [87]. There is also a positive correlation between serum leptin and estradiol concentration in postmenopausal women [88]. These positive correlations are supported by the investigations that reveal leptin enhancing aromatase expression and thus estradiol level through promoting nuclear translocation of AMP-responsive element binding protein-regulated transcription coactivator 2 (CRTC2) and decreasing the expression and activity of the AMP-activated protein kinase LKB1 [89,90]. Leptin itself has proinflammatory properties, by inducing macrophages and T cells to release inflammatory cytokines TNF-α, IL-6, and IL-17 [91] and enhancing endothelial COX-2 expression through p38 MAPK- and PI3K/AKT-activated mechanisms [92]. Therefore, apart from direct regulation of estrogen synthesis via aromatase, leptin probably increases estrogen production in an inflammatory process-dependent way. Reciprocally, enhanced production of estrogen and proinflammatory factors TNF-α and IL-1β induces leptin and its receptor in the adipose tissue, creating a feedback loop whereby leptin, proinflammatory factors, and estrogen amplify the effect of each other and thus result in an escalating inflammation and estrogen signaling [93,94]. In contrast with leptin, adiponectin consistently shows anti-inflammatory effects in a number of studies, converging at the mechanism that adiponectin suppresses the production of TNF-α and IFN*γ* by negatively regulating macrophages and dendritic cells and inhibiting NF-κB activation [95,96,97]. While the expression and secretion of adiponectin are inhibited by TNFα and IL-6 as a counteraction from proinflammatory components [98], it is interesting to find that estradiol suppresses adiponectin but not leptin expression in 3T3-L1 adipocyte cell line [99] and that serum adiponectin level is negatively correlated with estradiol level in postmenopausal women, thus bringing up the hypothesis that estrogen may antagonize the anti-inflammatory effect of adiponectin [100]. Taken together, this explains the clinical observation that the increased levels of leptin and decreased adiponectin secretion are directly associated with breast cancer development [101].

### 3.2. Estrogen in Breast Cancer and the Involvement of Inflammation

The role of inflammation as an instigator in tumorigenesis has been well acknowledged, and breast cancer is no exception [13]. As discussed in the previous section, the collective effect of inflammatory cytokines and mediators on aromatase is either enhancing expression or increasing activity, or even both, resulting in elevated estrogen levels in the breast. As estrogen is pro-proliferative and fuels tumor growth in the breast, the inflammation–aromatase–estrogen axis is postulated to drive tumor progression, especially in postmenopausal women where the in situ synthesized estrogen in the breast outweighs ovary-produced estrogen [102]. In fact, this axis is unequivocally manifested in research about the impact of obesity on breast cancer. While inflammation is a recognized phenotype of obesity [103], different meta-analyses have demonstrated the correlation between obesity (indicated by higher body mass index, BMI), estrogen production, and increased relative incidence of breast cancer in postmenopausal women and higher mortality risk of breast cancer in both pre- and postmenopausal breast cancer [104,105]. A more specific study on the gene expression of biomarkers in breast cancer revealed that the expression levels of IL-6, TNF-α, and leptin were higher in the breast adipose tissues of women with high waist circumference [106]. The high waist circumference approximates the accumulation of abdominal fat in obese women who have proportionally increased expression levels of cytochrome P450 family 19 subfamily A member 1 (CYP19A1), allograft inflammatory factor 1 (AIF1), and proinflammatory COX-2 [106]. The association of the inflammation–estrogen axis with breast cancer has been also validated in studies irrespective of obesity. Curtis TCGA data mining inversely correlated expression levels of oncostatin M (OSM), IL-6, and IL-1β with breast cancer patient survival [107]. Elevated serum inflammatory biomarkers including C-reactive protein (CRP), IL-6, and serum amyloid A (SAA) were positively associated with an increased risk of recurrence of HR+/HER2- tumor, and higher CRP level was correlated with detectable serum estrogen metabolites [108]. Intriguingly, IL-6 was positively correlated with prognostic factors in both ER+ and ER– patients [109]. The promoter polymorphism IL-6 may underlie the positive correlation of IL-6 with ER– breast cancer. On a molecular basis, IL-1β was found to induce the onset of epithelial–mesenchymal transition in breast cancer cells via IL-1β/IL-1RI/β-catenin signaling, which leads to the methylation of ERα gene promoter and the consequent ERα downregulation and tamoxifen resistance in a PI3K/AKT/TWIST-dependent way [110]. In addition, urinary metabolite level of PEG2, an inflammatory mediator that predicts estrone, was associated with an increased breast cancer risk [111].

While the interplay between inflammation and estrogen converges on aromatase in ER-positive breast cancers, the investigations into ER-negative breast cancers have expanded the knowledge about the conspiracies among inflammation and estrogen. Ovariectomy prevents the formation of not only ER-positive but also ER-negative breast cancers, suggesting that ovarian hormones are indispensable for ER-negative breast cancers [112]. The contribution of estrogen is apparent in promoting the growth of ER-negative breast cancers via a strong stromal effect rather than by directly working on cancer cells [112]. In the same study, estrogen was found to recruit bone marrow-derived cells to the mammary gland, and those cells were pro-proliferative. Later, the components of relocated myeloid cells were depicted. Macrophage infiltration and activation were facilitated by estradiol-elevated levels of chemoattractant CCL2 and CCL5 [113]. Neutrophil infiltration was also promoted by estrogen via enhancing CXCR2 signaling, and this population of neutrophils is polarized to be immunosuppressive and pro-tumorigenesis. Estrogen accelerated ER-negative 4T1 tumor growth in the mouse model, and this effect was ablated by neutrophil or granulocytic myeloid-derived suppressor cells (gMDSC) depletion [83,84,114,115]. It was also found that the alterations in inflammatory gene signature induced by estradiol were mediated by neutrophils [83,84]. Coincidentally, the indispensability of inflammatory components in ER-negative breast cancer was substantiated in a recent study showing that oncostatin M (OSM) and IL-1β synergistically induced IL-6 secretion in ER-negative MDA-MB-231 cell line to a greater extent than in ER-positive MCF-7 cell line [107]. These studies thus bridged the knowledge gap concerning estrogen crosstalk with inflammation components at a cellular level.

### 3.3. Anti-Inflammatory Role of Estrogens

The effect of estrogen on inflammation is complex and can be suppressive based on different pathophysiological conditions. For example, in ovariectomized mice mimicking the postmenopausal state, supplemental estrogen protected against high-fat-diet-induced white adipose tissue inflammation in the mammary gland and reduced the expression of proinflammatory factors TNFα, IL1β, COX-2, and aromatase in an ERα-dependent way [102]. This may explain the inverse or lack of correlation between excess body fat—which entails higher estrogen level—and breast cancer risk in premenopausal women [116]. It has been shown that at higher levels, E2 inhibits the expression of important proinflammatory cytokines such as TNFα, IL-6, IL-1β, MCP-1, and MMPs from both innate and adaptive immune cells, and at the same time, induces anti-inflammatory cytokines such as IL-4, IL-10, and TGF-β. At lower concentrations, however, E2 stimulates proinflammatory cytokines such as TNF, IFN-γ, and IL-1β. This concentration-dependent and biphasic role of E2 has been observed in T cells, dendritic cells, macrophages, and natural killer cells [9]. Specific anti-inflammatory effects of estrogen in breast cancer cells have also been reported. Exogenous expression of ERα has been found to reduce IL-8 secretion in breast cancer cells [117]. In MCF-7 breast cancer cells, E2 stimulates the expression of TGF-β [118], and inhibits MCP-1 production in a dose-dependent manner [119]. In human breast cancer samples, an inverse relationship between ER expression levels and macrophages has been established by a study [120]. ER-positive tumors exhibited a lower macrophage count. Moreover, ER-positive breast cancers have the lowest number of tumor-infiltrating lymphocytes (TILs), while TNBCs (that lack ER) generally have a greater TIL and inflammatory cell infiltrate compared to other breast cancer subtypes [121]. Consistently, an immune genetic profiling study found a negative correlation between TILs and *ESR1*/*ESR2* expression ratio in luminal breast cancer [122]. In the course of writing our review, we analyzed the expression of *ESR1* (the ERα gene) relative to that of several key inflammatory genes and CD molecules in the TCGA breast cancer dataset (N = 1247, results summarized in Table 1). Correlation studies were performed using built-in function of GraphPad Prism (9.2.0) on 628 luminal A or B breast cancers and revealed that ERα is in general negatively correlated with all inflammatory markers or immune cell infiltrations (as indicated by a Pearson correlation. The negative correlation is consistent with clinical observations that TNBCs tend to have increased immune cell infiltrations relative to ER-positive luminal breast cancers, one of the prerequisites for TNBC to have better response rate to immune checkpoint inhibitors (ICIs). The negative correlations between *ESR1* and inflammatory cytokines/immune cells support a role of ER-mediated signaling in immune suppression. In addition to the contrasting activities of estrogen via the direct impact on breast cancer cells, another recently discovered mechanism of estrogen on immune suppression is through the induction of myeloid-derived suppressor cells (MDSC)—an immune-suppressive cell type within the tumor microenvironment [114,115]. The estrogen-mediated inflammation/immune modulation provides the basis for rational design of potential therapeutic regimens, a point that will be discussed later in Section 5.

### 3.4. Endocrine Therapy for Breast Cancer and Inflammation

Informed with the pro-proliferative role of estrogen in the breast and its fueling effect in tumors, it is rational to reckon endocrine/antiestrogen therapy as the mainstay treatment for ER-positive breast cancer. Endocrine therapy can be used before surgery as a neoadjuvant treatment to downstage the disease and reduce the extent of surgery [123]. More often it is used as adjuvant therapy after surgery to reduce of risk of tumor recurrence [124]. Antiestrogens act through either lowering estrogen level or blocking estrogen from activating ER. The first category comprises aromatase inhibitors (AIs), namely, letrozole, anastrozole, and exemestane. AIs block the last step of estrogen biosynthesis, thus significantly reduce estrogen levels in postmenopausal women in whom estrogen is mainly converted from androgens by aromatase in peripheral tissues [125,126]. Tamoxifen and fulvestrant fall into the second category with different mechanisms of action. Tamoxifen and its analogues raloxifene and toremifene are called selective estrogen receptor modulators (SERMs) because they compete with estrogen to bind to ER and then exert mixed antiestrogenic and estrogenic actions on the ER signaling in a tissue- and cell-dependent manner [127]. In the breast, SERMs are clearly antiproliferative and pro-apoptotic, and hence counteract with the oncogenic effect of estrogen [128]. In the central nervous system and reproductive organs, they also demonstrate antiestrogenic features that account for most of the adverse effects of SERMs [129], but in bones and the liver, SERMs were evidenced to be estrogenic as patients receiving SERM therapy benefited from reduced levels of cholesterol and low-density lipoprotein and preserved bone density [130,131]. In contrast, selective estrogen receptor degraders (SERDs), prototyped by fulvestrant, show antiestrogenic effect throughout the body. SERDs not only completely antagonize ER but also degrade ER [132]; therefore, SERDs can circumvent the disadvantage of SERMs in that ER level can be upregulated as a feedback of the suppression of ER signaling by SERMs, thus decreasing the therapeutic efficacy of SERMs. Fulvestrant has proved its merit in postmenopausal women with disease progression after failure of other therapies [133]. As SERMs and SERDs both target ERα, the challenges poised to these agents are the mutation of ERα (much less impactful to SERDs than to SERMs) and the off-target activation of GPER, the two potential mechanisms accountable for endocrine resistance [134].

As tamoxifen is the mostly prescribed agent for endocrine therapy, we will use tamoxifen as an example to illustrate the impact of endocrine therapy in inflammation/immune modulation in addition to its direct inhibition of ER-positive cancer cells. Both ERs are expressed by most immune cell types, including T, B, and natural killer (NK) cells and myeloid lineages [135,136]. The impact of estrogen on immune cell subtypes has been summarized before [9,137]. Tamoxifen has recently been proposed as cancer immunotherapeutic [138] due to its direct impact on MDSC cells or tumor-associated macrophages via antagonizing estrogen signaling [114,115,139], which leads to increased T cell inflammation but reduced cancer cell-specific T cell activation and memory [115]. A phase II clinical trial (ClinicalTrials.gov, accessed on 5 November 2021, Identifier: NCT03879174) has been ongoing to determine the therapeutic responses of pembrolizumab (anti-PD-1) and tamoxifen among women with advanced ER-positive breast cancer. A case report confirmed the potential beneficial effect of this combination to treat ER-positive breast cancer [140]. The inflammation/immune modulation of other SERMs or SERDs was recently summarized [137] and will not be detailed here.

## 4. Xenoestrogens in Inflammation and Breast Cancer

Xenoestrogens occur naturally in plants or are chemically synthesized to manufacture consumer goods. These exogenous estrogens directly bind to the ligand-binding domain or other allosteric regions within ER primarily by their structural resemblance to the endogenous estrogens [141,142]. Xenoestrogens exhibit varying affinities to different estrogen receptors identified in humans. Of note, the functional activity of xenoestrogens is mediated by genomic and nongenomic pathways that involve various transcription factors, co-activators, and co-repressors, as well as interactions with other signaling pathways [143].

Humans are being continuously exposed to both natural and synthetic xenoestrogens. A significant amount of research has investigated how xenoestrogens alter physiological estrogen signaling in the human body and whether that promotes or prevents severe pathological conditions of estrogen-responsive tissues. Indeed, chronic exposure to synthetic xenoestrogens is considered a high-risk factor for the carcinogenesis of breast and other tissues. On the contrary, plant-derived xenoestrogens (aka phytoestrogens) have been shown to possess anti-inflammatory properties and are thought to be chemoprotective against breast cancer development. In this section, we discuss the regulation of inflammatory processes by phytoestrogens and synthetic xenoestrogens, and their significance in breast cancer pathogenesis.

### 4.1. Phytoestrogens and Cancer

Phytoestrogens, also referred to as plant-derived xenoestrogens or dietary estrogens, are polyphenolic or nonsteroidal compounds, naturally synthesized as secondary metabolites in plants [141,144]. Phytoestrogens possess a weak estrogenic activity and interfere with physiological estrogen signaling [145]. In the human gut, phytoestrogens are metabolized into active polyphenolic compounds capable of binding ERs with different affinities [146,147]. Depending on the circulating concentration of the endogenous estrogens and ER, phytoestrogens can evoke estrogenic or antiestrogenic properties that help maintain homeostatic estrogen signaling during estrogen deficiency or when there is overexpression of ER [141,148].

Four main classes of phytoestrogens frequently present in our diet: isoflavonoids and flavonoids (examples: genistein, daidzein, and quercetin; sources: legumes such as soy, chickpeas, and mung beans), lignans (examples: enterolactone and pinoresinol; sources: linseed, flaxseed, grains, and vegetables), stilbenes (example: resveratrol; sources: peanuts, pistachios, grapes, and blueberries), and coumestans (example: coumestrol; sources: alfalfa, clover sprouts, and sprouted legumes) [147]. Some commonly ingested and extensively studied phytoestrogens include genistein, daidzein, resveratrol, and quercetin.

Epidemiological data suggest that frequent consumption of phytoestrogens could be chemoprotective in breast, prostate, and colon cancers [144]. Daily intake of isoflavone-rich soy products is high in Asian countries such as China, Japan, Taiwan, and Korea. Studies have shown that women from these countries are at lower risk of developing estrogen-dependent cancers (including that of the breast) compared with their counterparts in the Western countries, where phytoestrogen consumption is relatively low [149,150]. Several case-control studies have shown significantly reduced breast cancer risk in women with a high intake of phytoestrogens such as isoflavones, lignan, and stilbenes [151,152,153]. Interestingly, many other studies have found small to no association between high phytoestrogen intake and reduced breast cancer incidence [153]. However, a few studies speculate that phytoestrogen intake must be high during prepubescent stages of development to achieve the protective benefits of phytoestrogens. In addition, another indirect impact of phytoestrogens is the alteration in the intestinal microbiota [154] that are known to regulate immune response and inflammation. Overall, findings from the epidemiological studies have been inconclusive, and as such, the chemoprotective role of phytoestrogens remains ambiguous.

Many phytoestrogens, including resveratrol, genistein, daidzein, and quercetin, have been demonstrated to bind both ERα and ERβ, albeit with much lower affinity than that shown by E2, and induce the transactivation of estrogen target genes in a dose-dependent manner [155,156,157,158]. Unlike E2, which binds ERα and ERβ with equal affinity, many phytoestrogens such as genistein and daidzein exhibit a remarkably higher binding affinity for ERβ than for ERα. Additional studies have shown that phytoestrogens, including genistein and resveratrol, display more robust transcriptional activity when bound to ERβ than when bound to ERα [142]. Provided the physiological concentration of phytoestrogens is just enough to activate ERβ but not ERα, due to their differential affinity, their net biological effect in the breast could be dictated by the ERβ-mediated signaling. Furthermore, decreased plasma concentration of endogenous estrogens, E1 and E2, has been reported in subjects with a high dietary intake of phytoestrogens, mainly flavonoids and isoflavonoids [159,160,161]. This observation is consistent with the findings that phytoestrogens can suppress the expression and activity of the metabolic enzymes, Cyp19 aromatase and 17β-hydroxy-steroid dehydrogenase (HSD), involved in estrogen biosynthesis [162,163].

In vitro and in vivo studies have extensively investigated the genomic and nongenomic activities of specific phytoestrogens, as well as their regulation of cell proliferation, survival, angiogenesis, and metastasis in various breast cancer models [145]. Many phytoestrogens, e.g., resveratrol, quercetin, and genistein, have been shown to exert a biphasic pattern in the regulation of breast cancer cell survival and proliferation in vitro, i.e., promote cell growth at low concentrations but inhibit cell growth and induce apoptosis at higher concentrations [142,147]. The antiproliferative and pro-apoptotic effects of phytoestrogens in breast cancer cells correlate with concomitant upregulation of tumor suppressors such as p21, p53, and p27 [164,165] as well as downregulation or reduced activation of oncogenes such as cyclin B1, Cdk1, and Bcl-2 [166,167]. Phytoestrogens such as genistein and resveratrol can also limit the activity of various protein tyrosine kinases, including PI3K and MAPKs, involved in breast cancer pathogenesis via nongenomic mechanisms. In vivo studies evaluating the antitumor or chemoprotective efficacy of phytoestrogens have yielded mixed results. Several groups have shown that genistein attenuates the growth of breast cancer xenografts in mice [168], whereas others found that it can also promote tumor growth [169]. In animal models of chemical-induced (e.g., dimethyl benz[a]anthracene (DMBA) or N-methyl-N-nitrosourea (NMU)) breast carcinogenesis, treatment with resveratrol, quercetin, genistein, or daidzein has been found to suppress breast cancer development [142,170,171]. However, in a few other instances, some of these phytoestrogens have been found to enhance tumor development [142]. Overall, an evaluation of results from such in vivo studies elucidates that the chemoprotective and antitumor roles of phytoestrogens in breast cancer are influenced by several factors, including the timing of phytoestrogen exposure, ERα vs. ERβ expression status of the tissue, and the local hormonal environment.

### 4.2. Role of Phytoestrogens in Inflammation

Inflammation can promote cancer progression, extracellular matrix degradation, invasion, and metastasis [172]. The use of nonsteroidal anti-inflammatory drugs (NSAIDs) may reduce the risk of developing cancer [173,174]. Mounting in vitro and in vivo evidence suggests that phytoestrogens have important anti-inflammatory functions that contribute to their chemoprotective role in breast cancer. Studies in laboratory animals and human subjects have shown that phytoestrogens can effectively reduce tissue inflammation by several mechanisms, discussed below [175,176] (as summarized in Table 2 and illustrated in Figure 3). 

#### 4.2.1. Reduced Expression and Activity of NF-κB and AP-1 Signaling

Many phytoestrogens such as resveratrol, genistein, and daidzein suppress the expression and transcriptional activity of nuclear factor-kappa B (NF-κB) and activator protein-1 (AP-1) in breast cancer cells in vitro and in vivo [177,178]. Inhibition of NF-κB and AP-1 has been well established to mediate the anti-inflammatory effects of resveratrol, genistein, and daidzein in mice [175,176]. Mechanistic studies revealed that such NF-κB and AP-1 downregulation or inactivation is mediated by ER-independent inhibition of the MEK, ERK, and p38 MAPK protein kinases by these phytoestrogens [179,180]. NF-κB is a key player of inflammation-associated cancers and regulates a multitude of genes regulating inflammation, e.g., cyclooxygenase-2 (COX-2), as well as cancer development and progression, e.g., c-Myc, p53, and MMP-9 [181]. Indeed, suppression of DMBA-induced mammary carcinogenesis by resveratrol correlates with the downregulation of NF-κB, COX-2, and MMP-9 in mice [177]. Moreover, in humans, the isoflavone genistein has been found to suppress tumor necrosis factor-α (TNF-α)-induced NF-κB activation in peripheral blood lymphocytes and limit oxidative DNA damage [182]. AP-1 is another transcription factor that regulates a wide range of cellular functions, including proliferation, differentiation, inflammation, and apoptosis [183]. AP-1 promotes the secretion of proinflammatory cytokines such as interleukin-8 (IL-8) as well as the proliferation of cancer cells [176]. In sum, current evidence suggests that suppression of NF-κB and AP-1 partly accounts for the antiproliferative and anticarcinogenic role of phytoestrogens in breast cancer. 

#### 4.2.2. Antioxidative Activity

Similar to endogenous estrogens, phytoestrogens upregulate key antioxidant enzymes such as Mn-SOD, catalase, and glutathione peroxidase, which scavenge mitochondria-generated ROS in cells [184,185,186]. Considering oxidative stress plays a major role in breast cancer etiology [187], the antioxidative activity of phytoestrogens is critical to impede tissue inflammation and ensuing mammary carcinogenesis [145,175]. Of note, genistein modulates oxidative stress in breast cancer cell lines according to ERα-to-β ratio, with its antioxidative effects observed only in breast cancer cells having low, but not high, ERα-to-β ratio [188]. This is consistent with our understanding that genistein and other phytoestrogens exhibit a remarkably higher affinity for ERβ, and as such are more reliant on ERβ-transduced signaling for their biological actions. In lipopolysaccharide (LPS)-challenged rats, oral administration of isoflavones and soy extracts reduces reactive nitrogen species such as nitrite, nitrate, and nitrotyrosine levels [189]. Furthermore, resveratrol has been shown to attenuate oxidative stress and inflammatory processes in rodents by augmenting Nrf2 and heme oxygenase-1 (HO-1), which constitute another key antioxidant pathway [190].

**Table 2 cancers-14-00206-t002:** Anti-inflammatory roles of phytoestrogens.

Mechanisms	Key Findings	References
Inhibition of NF-κB and AP-1 signaling	Suppression of DMBA-induced mammary carcinogenesis by resveratrol correlates with the inhibition of NF-κB	[177]
	Isoflavone genistein suppresses TNF-α-induced NF-κB activation in peripheral blood lymphocytes and oxidative DNA damage	[182]
	Genistein inhibits LPS-induced NF-κB activation in RAW 264.7 macrophages	[191]
Antioxidative activity	Genistein reduces oxidative stress in breast cancer cell with low ERα-to-ERβ ratio	[188]
	Isoflavones reduce reactive nitrogen species in LPS-challenged rats	[189]
	Resveratrol suppresses oxidative stress and inflammatory response related to hypoxic-ischemic brain injury in rats via Nrf2/HO-1 pathway	[190]
Reduced proinflammatory cytokine and chemokine generation	Isoflavones, including genistein, suppress the LPS-stimulated overproduction of IL-6, TNF-α, and IL-1β	[191,192,193]
	Isoflavone-rich diet reduces proinflammatory cytokines and immunosuppressive cells in pancreatic cancer patients	[194]
	Resveratrol attenuates lymphocytic IL-2 and IFN-γ production, as well as macrophageal IL-1β, IL-6, and TNF-α production	[195]
Suppressed COX-2 activity	Resveratrol attenuates DMBA-induced mammary carcinogenesis, which correlates with suppression of COX-2	[177]
	Resveratrol reduces COX-2 expression and activity in PMA-treated mammary epithelial cells	[196]
	Genistein and daidzein attenuates PMA-induced COX-2 expression in MCF-7 breast cancer cells	[197]
	Resveratrol suppresses lung and colorectal cancer cell proliferation via COX-2 downregulation	[198,199]

#### 4.2.3. Suppressed Proinflammatory Cytokines and Chemokine Production

Inhibition of proinflammatory cytokine and chemokine production is one of the major contributors of the phytoestrogen-induced anti-inflammatory effects [175,176]. Evidently, the genistein and daidzein isoflavones limit the production and release of macrophage- and monocyte-derived cytokines such as IL-1β, IL-6, IL-8, IL-12, and TNF-α in vitro and in vivo [175]. Interestingly, consumption of a soy isoflavone-rich diet in prostate cancer patients has been shown to reduce proinflammatory cytokines and immunosuppressive cells [194]. Resveratrol effectively suppresses the production of lymphocytic IL-2 and IFN-γ, as well as macrophageal TNF-α and IL-12 in a dose-dependent manner [195]. As many proinflammatory cytokines, including IL-1β, IL-6, and TNF-α, have been strongly linked to breast cancer progression [200], their suppression could be a key mechanism mediating the chemoprotective role of phytoestrogens in this cancer.

#### 4.2.4. Reduced Cyclooxygenase-2 Activity

Cyclooxygenase-2 (COX-2) catalyzes the conversion of arachidonic acid to PGH2, which is the precursor of several proinflammatory mediators, including prostaglandins, prostacyclin, and thromboxanes. COX-2 activation is at the mainstay of tissue inflammation induced by agents such as ultraviolet B, TPA (12-O-tetradecanoylphorbol-13-acetate), and PMA (phorbol 12-myristate 13-acetate), and tobacco [201,202], all of which are proven carcinogens. COX-2 is a transcriptional target of the NF-κB and AP-1 transcription factors, both of which are repressed by phytoestrogens. Consistently, phytoestrogens have been demonstrated to attenuate the expression and activity of the COX-2 enzyme in various contexts [148,176]. A study showed that resveratrol reduces prostaglandin E2 synthesis by directly limiting COX-2 expression and enzymatic activity in PMA-treated mammary epithelial cells [196]. In MCF-7 breast cancer cells, genistein and daidzein attenuate PMA-induced COX-2 transcription in an AP-1-dependent manner [197]. Furthermore, resveratrol has been shown to reduce lung cancer and colorectal cancer cell proliferation by downregulating COX-2 [198,199]. Taken together, COX-2 suppression is one of the important mechanisms by which phytoestrogens exert their anti-inflammatory and antitumor functions.

Although the majority of studies have evidenced the anti-inflammatory and chemoprotective role of phytoestrogens, some contradictory findings have been made. Many phytoestrogens, including genistein, have been found to suppress humoral and cell-mediated immunity in mice via ER-dependent or -independent pathways [203,204]. An epidemiological study in Hawaii associated long-term consumption of a soy-rich diet with the etiology of Kawasaki disease, which is marked by inflammation of the blood vessels [205]. Moreover, sustained exposure to genistein has been found to accelerate the growth of breast cancer xenografts in mice and tumor progression [206]. Several compounds, including genistein, resveratrol, and xanthohumol, can paradoxically exert oxidative effects that might promote an inflammatory reaction [145]. Such observations highlight the need to use phytoestrogens with caution and to perform more studies to evaluate the true potential of phytoestrogens in cancer prevention and therapy.

### 4.3. Synthetic Xenoestrogens and Cancer

Synthetic xenoestrogens (simply referred to as xenoestrogens) comprise a subset of endocrine-disrupting chemicals that can directly interact with ERs or modulate ER signaling via indirect mechanisms [207]. Many xenoestrogens also dysregulate the metabolism of endogenous estrogens, causing an increase or decrease in their physiological concentration [208]. Xenoestrogens might adversely impact the growth and physiology of estrogen-sensitive organs, including the breast. Chemically, xenoestrogens are diverse compounds synthesized for use in agriculture, industry, and consumer goods. Bisphenol A (BPA), dichloro-diphenyl-trichloroethane (DDT), polychlorinated biphenyls (PCBs), polycyclic aromatic hydrocarbons (PAHs), and 2,3,7,8-tetrachlorodibenzo-p-dioxin (TCDD, dioxin) are some common synthetic xenoestrogens that humans are frequently exposed to.

Many epidemiological data have associated an increased rate of breast cancer incidence in women to rising human exposure to specific environmental xenoestrogens such as DDT [208,209,210,211]. Consistently, experimental data obtained from a large number of in vitro and in vivo studies have demonstrated the carcinogenic potential of xenoestrogens in the breast [208,212]. It is noteworthy that concentrations required to induce the carcinogenic effects in in vitro and in vivo models are exceptionally higher than what humans are normally exposed to [147]. Moreover, many additional epidemiological and experimental studies have found no association between such xenoestrogens and carcinogenesis of the breast in women [208,209]. Considering such inconsistent discoveries in the field, whether specific xenoestrogens enhance the risk of breast cancer is still a debatable topic [209,211].

Breast tissue in women undergoes intense morphological changes and cell proliferation during prenatal development, puberty, pregnancy, and the menopausal transition. In these specific susceptibility windows, the breast epithelium is highly sensitive to estrogenic compounds, so that women exposed to certain xenoestrogens during these periods are at increased risk of breast cancer development [209,211]. For example, women exposed to diethylstilbestrol (DES) in utero are highly susceptible to developing breast cancer after 40 years of age [213]. Animal studies have shown that prenatal exposure to 2,3,7,8-tetrachlorodibenzo-p-dioxin (TCDD, dioxin) enhances susceptibility to carcinogen-induced breast cancer development, while exposure during pregnancy delays the tumor formation [214]. In sum, the role of xenoestrogens in modulating mammary carcinogenesis, if any, could be highly dependent on the timing of exposure and the status of target cells.

Synthetic xenoestrogens such as BPA and DDT exhibit very weak affinity for both ERα and ERβ [147,212,215]. Considering their serum concentration in humans is at least 1000-fold lower than what is required to half-saturate ER, it is greatly unexpected for them to have any significant estrogenic impact on target tissues in vivo [216,217,218,219]. However, xenoestrogens are very lipophilic compounds, which can bioaccumulate and persist in fatty tissues of the body for years. Indeed, their concentration in fatty tissues of the human body was found to be nearly 1000-fold higher than the serum levels, which potentially evokes estrogenic or antiestrogenic effects in those target tissues. Other mechanisms by which xenoestrogens can modulate estrogen signaling include regulation of ER expression, signal transduction from other receptors, epigenetic alterations, as well as estrogen metabolism and transport [220].

### 4.4. Specific Synthetic Xenoestrogens in Inflammation and Breast Cancer

Chronic exposure to synthetic xenoestrogens could promote mammary carcinogenesis by inducing local tissue inflammation [207]. Besides affecting cell transformation, proliferation, and apoptosis, xenoestrogens have been found to alter the local immune microenvironment of the target tissue, which can trigger inflammation [221]. Although acute inflammation can be reparative, chronic inflammation is often associated with a variety of pathological conditions, including cancer [172]. In the following section, we discuss the role of important xenoestrogens in breast cancer pathogenesis with a special focus on their inflammatory actions (summarized in Table 3 and illustrated in Figure 3).

#### 4.4.1. Bisphenol A (BPA)

BPA is extensively used as a plasticizer in the manufacturing of polycarbonate plastics and epoxy resins [221]. Major sources of human exposure to BPA are drinking water in highly industrialized areas as well as its leaching from plastic-lined food, drinks, and dental sealants. Several epidemiological studies have reported a significant correlation between the urinary concentration of BPA (indicative of BPA exposure) and high breast density, which is a high-risk factor for breast cancer [222,223]. However, other studies have found no association [224,225,226]. Moreover, BPA has been demonstrated to induce neoplastic transformation of human breast epithelial cells in vitro [212] and to promote mammary carcinogenesis in rodents following their in utero exposure [227,228].

BPA exposure can affect the differentiation, proliferation, and secretion of various immune cell types that govern tissue inflammation. BPA has been shown to induce the differentiation of CD4+ T lymphocytes into either the proinflammatory Th1 or Th17 subsets or into the anti-inflammatory Th2 subsets [221]. Additionally, BPA exposure can enhance T cell proliferation, which may trigger an inflammatory disease. BPA exposure in prenatal or adult mice was found to reduce the number of regulatory T cells, with a greater effect observed in the prenatal exposure group [229]. Moreover, oral BPA administration can also increase the number and immunoglobulin production of B cells in mice [230]. Several in vitro studies have revealed that BPA treatment can either suppress [231] or enhance [232] LPS-induced production of NO in a concentration-dependent manner. Such proinflammatory effects of BPA on immune cells are mediated by its activity on ER, aryl hydrocarbon receptor (AhR), or peroxisome proliferator-activated receptor (PPAR) [221].

**Table 3 cancers-14-00206-t003:** Effects of synthetic xenoestrogens on inflammation and immune system.

Xenoestrogen	Key Findings	References
Bisphenol A (BPA)	Induces CD4+ T lymphocyte differentiation into the proinflammatory Th1 or Th17 subsets	[233,234,235]
	Induces CD4+ T lymphocyte differentiation into the anti-inflammatory Th2 subsets	[236,237,238]
	Reduces regulatory T cell number in prenatal and adult mice	[229]
	Increases the number and immunoglobulin production of B cells in vivo	[230]
	Either suppresses or enhances LPS-induced NO production	[231,232]
Dichloro-diphenyl-trichloroethane (DDT)	Upregulates COX-2 and prostaglandins in breast cancer cells	[239]
	Elevates inflammatory and oxidative stress marker genes, *CXCL8*, *HMO-1*, and *TNF,* in MCF-7 breast cancer cells	[240]
	Induces oxidative DNA damage culminating in hepatic neoplasia	[241]
	Suppresses antigen-induced serum γ-globulin levels	[242]
	Upregulates inducible iNOS and proinflammatory cytokines in NF-κB-dependent manner	[243]
Polychlorinated biphenyls (PCBs)	PCB 126 induces macrophage polarization to the proinflammatory M1 and enhances the secretion of TNF-α and IL-1β. It also induces oxidative stress marker genes.	[244]
	PCB 126 disrupts gut microbiota, as well as promotes intestinal and systemic inflammation	[245]
	PCB 153 promotes NF-κB-mediated inflammation and oxidative stress	[246]
	Implicated in immunosuppression and delayed immune response	[247,248]
Polycyclic aromatic hydrocarbons (PAHs)	Induces oxidative stress	[249,250]
	Elicits serious immunotoxic and immunosuppressive effects in mice	[251,252,253]
	Promotes Th17 differentiation of T lymphocytes and increases IFN-γ-positive dendritic cell population	[254]
2,3,7,8-Tetrachlorodibenzo-p-dioxin (TCDD)	Induces inflammatory cytokine production by dendritic cells and macrophages Elicits immunosuppressive effects in mice by disrupting peripheral T lymphocyte development	[255]
	Activates AhR to increase ROS generation and proteolytic maturation of IL-1β	[256]

#### 4.4.2. Dichloro-Diphenyl-Trichloroethane (DDT)

DDT is an organochlorine pesticide, classified as “probably carcinogenic to humans” by the International Agency for Research on Cancer (IARC) [209]. Once absorbed, DDT is converted to its active metabolite, DDE, which is highly lipophilic and capable of bioaccumulating in the body tissue and fluids. In 1993, Wolff et al. reported a significant association between DDT exposure and breast cancer risk in women [219]. Since then, hundreds of epidemiological studies have investigated the association of DDT exposure with breast cancer risk in women. Many of the large cohort studies, considering the particular windows of susceptibility in women, strongly link childhood exposure to DDT with increased risk of breast cancer [209]. However, several other studies have found no association between these two, inviting some level of uncertainty as to the carcinogenic role of DDT in the breast [211]. Nevertheless, experimental studies have demonstrated strong tumor-promoting effects of DDT using in vitro and in vivo models of breast cancer [208,209,257,258].

The proinflammatory role of DDT has been illustrated by many investigations. A study showed that DDT treatment upregulates COX-2 and prostaglandins in breast cancer cells [239]. MCF-7 breast cancer cells exposed to environmentally relevant concentrations of DDT exhibit upregulation of inflammatory and oxidative stress marker genes such as *CXCL8, HMOX-1,* and *TNF* [240]. In rodents, DDT ingestion has been shown to induce oxidative DNA damage resulting in hepatocellular eosinophilic foci and hepatic neoplasia [241]. DDT also possesses immunosuppressive actions in that continued dietary DDT treatment in rabbits suppresses their antigen-induced serum γ-globulin levels [242]. Moreover, treatment of macrophages with DDT has also been demonstrated to upregulate inducible nitric oxide synthase (ins) and proinflammatory cytokines in an NF-κB-dependent manner [243].

#### 4.4.3. Polychlorinated Biphenyls (PCBs)

PCBs are non-inflammable, chemically stable compounds used in plasticizers, pigments, capacitors, and transformers. Like DDT, PCBs have been declared ‘probably carcinogenic to humans’ by the IARC, requiring their highly restricted usage. Elevated breast cancer rates have been reported in women exposed to PCBs [219,259,260], whereas a few other studies have found no significant correlation between PCB exposure and breast cancer incidence [208,261]. Nonetheless, the carcinogenic role of specific PCBs in breast cancer has been established by multiple in vitro and in vivo studies [207].

PCBs can precipitate tissue inflammation via several mechanisms. PCB 126 induces macrophage/monocyte polarization to the proinflammatory M1 phenotype, and as such, enhances the secretion of TNFα and IL-1β in an AhR- and NF-κB-dependent fashion [244]. Furthermore, PCB 126 can induce oxidative stress in cells, as evidenced by the upregulation of oxidative stress marker genes such as *NRF2*, *HMOX1*, and *NQO1*. In mice, PCB 126 administration disrupts gut microbiota as well as promotes intestinal and systemic inflammation [245]. Other PCBs have also been shown to promote NF-kB-mediated inflammation and oxidative stress, resulting in glucose/lipid metabolic disorder in mice [246]. Additionally, PCB exposure has been implicated in immunosuppression and delayed immune response in both rodents and humans [247,248].

#### 4.4.4. Polycyclic Aromatic Hydrocarbons (PAHs)

PAHs are a large group of organic compounds containing two to seven fused benzene rings [262]. Several PAHs such as benz[a]anthracene, benzo[a]pyrene, dibenz[a,h]anthracene, and DMBA are well-established human carcinogens. PAHs are released into the environment from the incomplete burning of organic substances, including fossil fuels. Epidemiological data suggest that chronic exposure to specific environmental PAHs is strongly associated with many cancers, including that of the breast in women [208]. Their carcinogenic properties have further been corroborated by many animal studies [208,263].

PAHs are classified as immunosuppressive agents that are likely to compromise body resistance against infections and cancer progression [264]. A significant correlation between carcinogenicity and immunotoxicity of PAHs has been observed [265]. PAHs and their oxidative metabolites can induce ROS production and serious genotoxic effects in immune cells. Immunotoxicity of PAHs has been noticed in bone marrow, thymus, spleen, and peripheral lymphoid tissues. Moreover, PAHs are moderate-to-strong AhR ligands. PAHs present in ambient urban dust promote the proinflammatory Th17 differentiation in T cells and increase the IFN-γ-positive dendritic cell population in an AhR-dependent manner [254]. Benzo[a]pyrene has been shown to decrease the count of lymphocytes, eosinophilic granulocytes, B cells, and natural killer cells as well as immunoglobulin levels in rats [253]. In MDA-MB-231 breast cancer cells, PAH exposure activates the NADPH oxidase enzyme, indicative of aggravated oxidative stress [250].

#### 4.4.5. 2,3,7,8-Tetrachlorodibenzo-p-Dioxin (TCDD)

TCDD is a serious human carcinogen according to the IARC [209]. Although TCDD use has been strictly regulated, some dioxin emission still arises from electric power plants, iron and steel industries, as well as nonferrous metal manufacturers [266]. Many epidemiological studies have shown a significant correlation between elevated serum TCDD concentration with a high risk of breast cancer in women [267,268]. Although no association between TCDD exposure and breast cancer risk was found by other studies [269,270], they reported a significant connection of TCDD with other malignancies such as biliary cancer, brain cancer, and lymphatic and hemopoietic neoplasm [261]. TCDD promotes breast cancer cell proliferation and chemoresistance in vitro [271,272]; whereas in utero, TCDD exposure in rodents enhances their susceptibility to breast cancer development [273,274].

Apart from modulating ER signaling, TCDD can also activate AhR (also known as dioxin receptor), leading to various inflammatory responses [209]. AhR is well expressed in dendritic cells and macrophages, which are important mediators of the innate immune system [275]. Several studies have shown that TCDD exposure induces inflammatory cytokine production by dendritic cells and macrophages [255]. TCDD-stimulated AhR activation also leads to increased ROS generation and proteolytic maturation of inflammatory cytokines such as IL-1β [256]. Furthermore, TCDD exposure has been shown to evoke immunosuppressive effects in mice by disrupting their peripheral T lymphocyte development [255].

## 5. Concluding Thoughts and Perspectives

Women at all ages are subject to the physiological and pathological effects of estrogens that are originated naturally or environmentally. The research articles reviewed here have delineated how natural and environmental estrogens participate in physiological processes as well as carcinogenesis. In sum, chronic and long-term activation of estrogen receptor is carcinogenic, which can provoke an inflammatory environment that generally supports the estrogen-related risk of tumorigenesis. Endogenous estrogen signaling pathway may be locally escalated by the elevated estrogen production from the ovary and/or the tumor microenvironment, or via mutations of estrogen receptors, a process that can switch from proinflammation to anti-inflammation, as discussed in Section 3.3. Endocrine therapy has been the mainstay regimen for treating ER-positive luminal breast cancers by specifically targeting ER-mediated signaling. For a long time, it was believed that endocrine therapy mainly induces apoptosis or suppresses the proliferation of cancer cells, until the benefit effects of these anti-ER therapeutics were revealed to inhibit the ER-negative cancers. The immune modulation by endocrine therapy can be intrinsic to cancer cells—via antagonizing the ER-mediated suppression of PD-L1 immune checkpoint [276]—or extrinsic to cancer cells via direct modulation of various immune cells [137]. The impact of endocrine therapy on immune cells can be either beneficial or detrimental to cancer treatment [137], depending on the dominant immune cell subtypes within the tumor microenvironment. For example, the induction of PD-L1 by tamoxifen/fulvestrant may benefit patients with the treatment of anti-PD-1/PD-L1 antibodies. The induction of Tregs by tamoxifen [277] may benefit patients with the treatment of anti-CTLA-4 (ipilimumab) or known Treg-depleting chemotherapeutics. The depletion of MDSC and subsequent activation of T cells [114,115] by tamoxifen may benefit patients with further immune checkpoint inhibitors. These considerations should be carefully considered when designing clinical trials in ER-positive breast cancer or other cancer types.

The innate estrogen signaling and inflammatory pathways constantly interact with both natural and synthetic xenoestrogens. However, the net biological effect of these interactions could be dictated by several factors, including their relative rates of exposure, affinities towards a specific ER or inflammatory modulator, metabolism, and bioaccumulation. Chronic exposure to synthetic xenoestrogens can cause serious adverse impacts through their action as endocrine disruptors or cancer promoters. Nevertheless, most of their biological effects are thought to be competitively blocked from ER sites by dietary phytoestrogens, as the latter are consumed at much higher quantities in common foods with higher binding affinity to ER [147]. However, xenoestrogens such as DDT and PCBs can bioaccumulate in fatty tissues of the body and exhibit sustained effects due to their very long biological half-lives, whereas phytoestrogens such as those found in soy foods are metabolized and eliminated within hours. This suggests regular consumption of a phytoestrogen-rich diet could be helpful to keep the deleterious effects of synthetic xenoestrogens in check.

There is still not enough evidence to support the clinical use of phytoestrogens in cancer therapy. The potential of phytoestrogens as anticancer agents has been extensively reviewed in literature [145,173]. Many phytoestrogens administered alone or in combination with established therapies have been shown to possess promising antitumor effects such as enhanced cell killing and averted drug resistance in various in vitro and in vivo cancer models. However, contradictory results showing no effect or interference with existing cancer treatments have also been observed in many instances. Moreover, previous phase I and II clinical trials have essentially found no significant therapeutic benefit of co-administering phytoestrogens in cancer patients undergoing surgery or some other forms of treatment [173]. The sample size in those studies was small, and treatment dosage and length were not adequately monitored. Encouragingly, high soy protein or isoflavone intake was associated with reduced breast cancer recurrence and mortality in a large population-based study in China [278]. Further large, controlled clinical studies are required to better evaluate the potential of phytoestrogens as an option in cancer therapy, establish the optimal dose, and identify which patients could benefit from them.

## Figures and Tables

**Figure 1 cancers-14-00206-f001:**
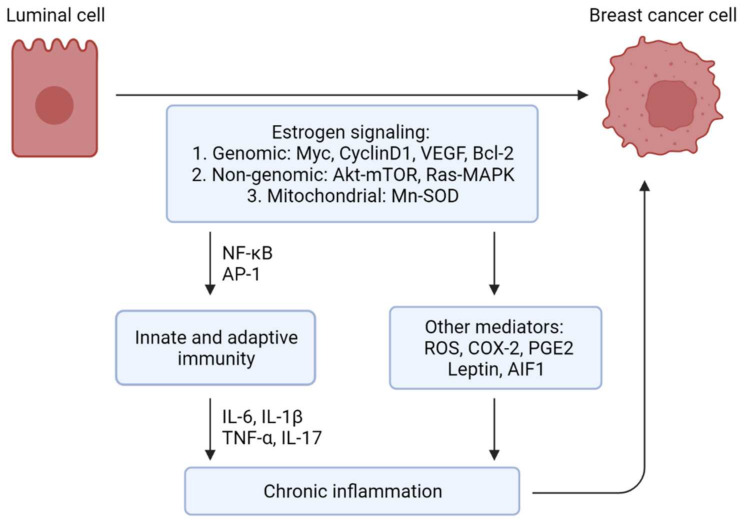
Schematic representation of estrogen signaling and downstream pathways regulating chronic inflammation and breast cancer etiology. Estrogen drives luminal epithelial cell transformation leading to breast cancer development via genomic, nongenomic, and mitochondrial signaling pathways. Besides promoting cell survival, proliferation, angiogenesis, and metastasis, these pathways can also modulate inflammation, innate and adaptive immunity by upregulating or activating proinflammatory mediators such as reactive oxygen species (ROS) and cyclooxygenase 2 (COX-2). Estrogen-induced chronic inflammation is an important co-conspirator in breast cancer development and progression. The anti-inflammatory function of ER signaling is also documented in pathophysiology but not discussed here. Key abbreviations: VEGF, vascular endothelial growth factor; MAPK, mitogen-activated protein kinase; Mn-SOD, Manganese superoxide dismutase; NF-κB, nuclear factor-kappa B; AP-1, activator protein-1; IL, interleukin; TNF-α, tumor necrosis factor-alpha.

**Figure 2 cancers-14-00206-f002:**
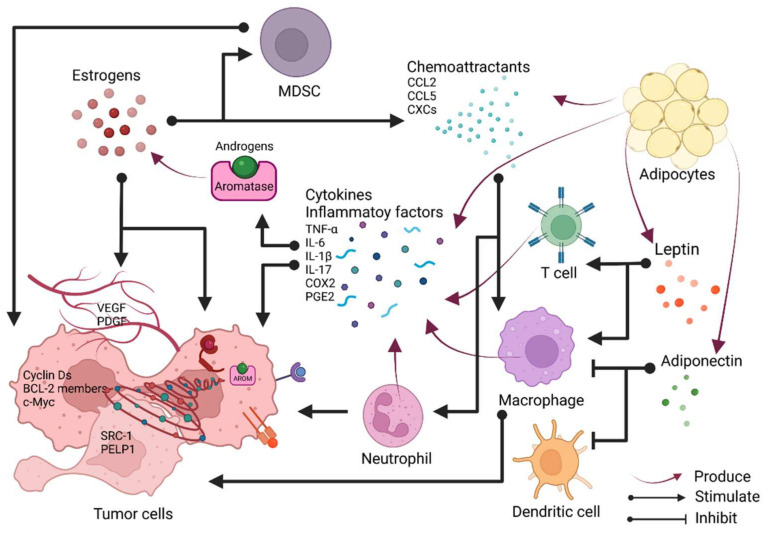
Schematic depiction of the tumor microenvironment of breast cancer. Tumor cells, infiltrated immune cells, and adipocytes are highlighted in this schematic graph. Estrogens drive tumor cell proliferation and metastasis via transcriptional activation of cyclin Ds, anti-apoptotic factors, SRC-1, and PELP1. Estrogens also promote angiogenesis via VEGF and PDGF. Cytokines such as TNFα, IL-6, andIL-1β and inflammatory components are released mainly by infiltrated immune cells. They increase the activity of aromatase and thus elevate the estrogen level, and they also directly impact the growth of tumor cells via receptors. Estrogens elevate chemoattractant levels and thus facilitate the infiltration of MDSC, neutrophil and macrophage which then promote tumor growth. Adipocytes secrete inflammatory factors and leptin, which stimulate T cells and macrophages to produce more cytokines, forming a positive feedback, while adiponectin can inhibit immune cells.

**Figure 3 cancers-14-00206-f003:**
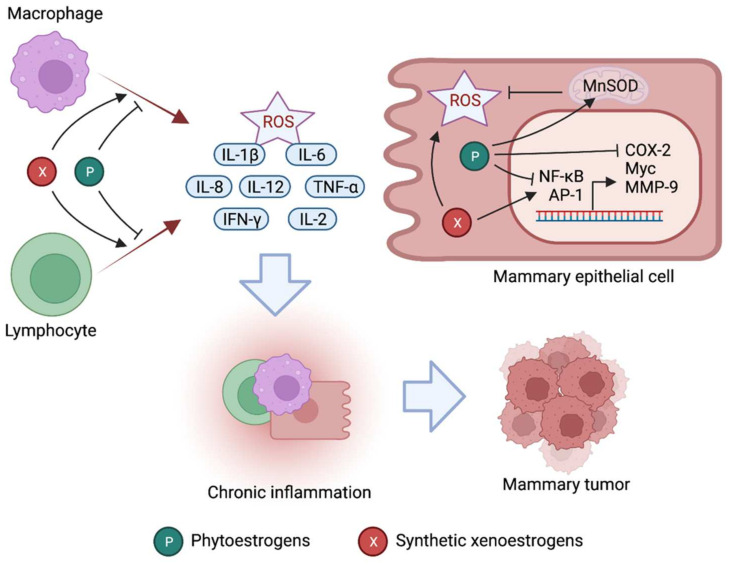
Inflammatory mechanisms differentially regulated by phytoestrogens and synthetic xenoestrogens. Phytoestrogens promote anti-inflammatory responses in mammary tissue via inhibition of NF-κβ and AP-1, reduction of reactive oxygen species (ROS) in a manganese superoxide dismutase (MnSOD)-dependent manner, inhibition of COX-2, and reduced secretion of proinflammatory cytokines from innate and adaptive immune cells. On the other hand, synthetic xenoestrogens generally induce proinflammatory responses by activating NF-κβ signaling, increasing ROS generation, and enhancing the proinflammatory cytokine release from immune cells. Such responses can propagate chronic inflammation leading to breast cancer development.

**Table 1 cancers-14-00206-t001:** Correlation of ERα expression with inflammatory cytokines. (* *p* < 0.05, statistically significant).

Inflammatory Marker	Pearson r	*p*-Values
TNF-α	−0.077	0.069
IL-6	−0.635	<0.0001 *
IL-1β	−0.084	0.060
IL-1α	−0.011	0.710
IL-17	−0.020	0.024 *
TGF-β	−0.171	<0.0001 *
CXCL-1	−0.484	<0.0001 *
Leptin	−0.704	<0.0001 *
**CD Molecules**	**Pearson r**	***p*-Values**
CD4	−0.319	<0.0001 *
CD5	−0.255	<0.0001 *
CD83	−0.270	<0.0001 *
CD163	−0.268	<0.0001 *
CD40	−0.421	<0.0001 *
CD34	−0.380	<0.0001 *

## Data Availability

Not applicable.

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
