# Peer review of "Natural and Synthetic Estrogens in Chronic Inflammation and Breast Cancer"

_cancers, 2021, doi:10.3390/cancers14010206_

Round 1

Reviewer 1 Report

This is a well written and well organized manuscript on an important topic. I consider this a highly relevant topic in the field, actually was thinking to write a review for the same topic!

The review analyzes the literature how natural or synthetic estrogens modulate chronic inflammation and breast cancer. The recent literature is well presented, and the topic will not only help to understand the influence of estrogens, but also to find better and more effective treatment options.

Having this said, it would be helpful to have an illustration of how the different natural or synthetic estrogens modulate the different pathways (resulting in inflammation or breast cancer). A summarizing figure(s) would be very helpful for the reader to see which pathways are targeted by estrogens and what are the possible consequences for the fate of the cell or how they could modulate inflammation!

Otherwise, this is a well done review.

Author Response

Thank you for the suggestion! A new schematic (Figure 3) showing the differential regulation of inflammatory pathways by natural and synthetic estrogens has been added.

We also include a new graphic abstract for overall review of the content.

Reviewer 2 Report

The review by Maharjan et al. focuses on the tumor-stimulating properties and more thoroughly on the inflammatory nature of estrogen.

Indeed, estrogen-induced chronic inflammation is known to be a significant contributor in breast cancer pathogenesis through upregulation of pro-inflammatory mediators such as reactive oxygen species (ROS) and cyclooxygenase (COX-2). The anti-inflammatory functions of estrogen mainly involve the reduction of T cell infiltration and production of immunosuppressive cytokines, including IL-4, IL-10 and TGF-beta.

Overall, the review provides a clear and extensive image of the current and relevant literature.

A few suggestions are in place to further enhance the strength and comprehensiveness of the manuscript:

1) A comprehensive figure showing a schematic overview of the (breast) tumor microenvironment and how estrogen impacts on the different stromal/tumor components could help the reader in understanding the most important findings/take-home message.

2) Given the differential impact of phytoestrogens and synthetic xenoestrogens, could the authors provide a clinical recommendation for use in patients?

3) It would be interesting to specify future challenges and opportunities that arise for anti-hormonal treatment based on the literature screening performed in this review.

4) As an additional minor comment, on page 7, line 333, there is a reference missing (citation?).

Author Response

A few suggestions are in place to further enhance the strength and comprehensiveness of the manuscript:

1) A comprehensive figure showing a schematic overview of the (breast) tumor microenvironment and how estrogen impacts on the different stromal/tumor components could help the reader in understanding the most important findings/take-home message.

A new Figure 2 is added.

2) Given the differential impact of phytoestrogens and synthetic xenoestrogens, could the authors provide a clinical recommendation for use in patients?

A discussion on the potential use of phytoestrogens in cancer therapy has been added in the conclusions section. In the preceding new paragraph, we have also talked about the interaction between phytoestrogens and synthetic xenoestrogens at various levels in the human body.

3) It would be interesting to specify future challenges and opportunities that arise for anti-hormonal treatment based on the literature screening performed in this review.

Discussion is added in section 5.

4) As an additional minor comment, on page 7, line 333, there is a reference missing (citation?).

Thanks for pointing this out. The citation has been added.